

# Habitat use, preference, and utilization distribution of two crane species (Genus: *Grus*) in Huize National Nature Reserve, Yunnan–Guizhou Plateau, China

Dejun Kong[1,2,*], Weixiong Luo[1,*], Qiang Liu[3], Zhuoqing Li[4], Guoyue Huan[5], Jianjun Zhang[5] and Xiaojun Yang[1]

[1] Kunming Institute of Zoology, Chinese Academy of Sciences, Kunming, Yunnan, China
[2] Key Laboratory of Special Biological Resource Development and Utilization of Universities in Yunnan Province, Kunming University, Kunming, Yunnan, China
[3] College of Wetland, Southwest Forestry University, Kunming, Yunnan, China
[4] Yunnan Institute of Environmental Science, Kunming, Yunnan, China
[5] Administrative Bureau, National Nature Reserve of Black-Necked Cranes in Huize, Huize, Yunnan, China
* These authors contributed equally to this work.

Corresponding author
Xiaojun Yang, yangxj@mail.kiz.ac.cn

## ABSTRACT

Understanding the habitat use and spatial distribution of wildlife can help conservationists determine high-priority areas and enhance conservation efforts. We studied the wintering habitat use, preference, and utilization distribution of two crane species, that is, the black-necked crane (*Grus nigricollis*, Przevalski, 1876) and common crane (*Grus grus*, Linnaeus, 1758), in Huize National Natural Reserve, Yunnan–Guizhou Plateau, southwestern China. Line transects indicated that anthropogenic farmland habitat was highly utilized and was positively selected by both crane species (>90% of flocks observed for both species). Black-necked cranes preferred marshland in spring (February and March) but avoided grassland during the entire wintering period, whereas common cranes avoided both marshland and grassland throughout the entire period. The two cranes species had communal nightly roosting sites and separate daily foraging sites. Black-necked cranes were distributed within two km (1.89 ± 0.08 km) of the roosting site, covering an area of 283.84 ha, with the core distribution area encompassing less than 100 ha. In contrast, common cranes were distributed far from the roosting site (4.38 ± 0.11 km), covering an area of 558.73 ha, with the core distribution area encompassing 224.81 ha. Thus, interspecies competition may have influenced the habitat preference and spatial distribution divergence of these two phylogenetically related species. This study should help guide habitat management as well as functional zoning development and adjustment in the future. Based on our results, we recommend restoration of additional wetlands, retention of large areas of farmland, and protection of areas that cranes use most frequently.

## INTRODUCTION

Understanding the habitat use and spatial distribution of wildlife is important for conservation and management (*Morris, 2003*; *Klar et al., 2008*). Habitat contains all the resources and conditions influencing the survival and reproduction of resident wildlife (*Block & Brennan, 1993*; *Odum & Barrett, 2004*). Effective conservation, especially of endangered species, needs a deep understanding of habitat and frequency of use and as well as its relationship with populations (*Block & Brennan, 1993*; *Jones, 2001*). By defining the relative frequency of occurrence of animals (utilization distribution), ecologists and conservationists can obtain a global representation of spatial use (*Benhamou & Riotte-Lambert, 2012*). Utilization distribution can help determine protection areas of high priority and highlight essential habitat management (*Cañadas et al., 2005*).

Black-necked cranes (*Grus nigricollis*, Przevalski, 1876) are characterized as vulnerable on the IUCN Red List of Threatened Species (*BLI, 2018*) and Biodiversity Red List of China (*MEP & CAS, 2015*). These cranes mainly inhabit the alpine wetlands of the Qinghai–Tibetan and Yunnan–Guizhou plateaus of China, with a total population of 10,000–10,200 individuals (*Li, 2014*). Nearly all breeding populations of black-necked cranes are distributed on the Qinghai–Tibetan Plateau, except for a small number (maximum 139 birds) in adjacent Ladakh, India (*Chandan et al., 2014*). Their wintering area includes lower elevations on the Qinghai–Tibetan and Yunnan–Guizhou plateaus, as well as in Bhutan and occasionally in Nepal, Myanmar, Vietnam, and Kashmir (*Li, 2014*; *Chandan et al., 2014*). This species is facing threats from habitat loss and degradation induced by anthropogenic activities and climate change, with human disturbances particularly serious in their wintering grounds (*Harris & Mirande, 2013*; *Li, 2014*). Despite this, population increases over the past thirty years are believed to have occurred due to the benefits of grain waste in farmlands during winter (*Harris & Mirande, 2013*). However, conflicting results on black-necked crane habitat use have been reported from different wintering sites on the Yunnan–Guizhou Plateau. For example, *Li (1999)* observed that 54.4%, 26.8%, 11.4%, and 7.3% of cranes from the Caohai Reserve, Guizhou Province, were distributed in sedge meadow, farmland, shallow marshland, and grassland, respectively. Conversely, *Liu et al. (2010)*, who studied the winter foraging habitat utilization of black-necked cranes in Napahai Reserve, southwestern Yunnan, indicated that 75.2% of cranes used shallow marshland, whereas only 6.7% of cranes were observed in farmland. However, *Kong et al. (2011)* reported that wintering black-necked cranes in Dashanbao Reserve, northeastern Yunnan, most often utilized farmland (55.1%) and concluded that landscape differences between wintering sites resulted in the observed differences in wintering habitat use. Thus, habitat preference, which can reflect the biological characteristics of an animal (*Hall, Krausman & Morrison, 1997*), should be considered in further studies. Habitat use refers to the way in which an individual or species uses habitat to meet its life history needs (*Jones, 2001*), whereas habitat preference also considers habitat availability, resulting in the disproportional use of some resources over others (*Krausman, 1999*). Both use and preference are consequences of habitat selection (*Block & Brennan, 1993*). However, the crane habitat preference studies mentioned above also

reported distinct results. The habitat preference rank of black-necked cranes in Caohai Reserve was sedge meadow > grassland > shallow marsh > farmland (*Li, 1999*), whereas the cranes in Dashanbao Reserve preferred shallow marshland and farmland and avoided grassland altogether (*Kong et al., 2011*). Thus, additional case studies on the habitat use and preference of black-necked cranes should be conducted in consideration of the contradictory results and the critical conservation of this bird species on the Yunnan–Guizhou Plateau.

Here, we studied the wintering habitat use, preference, and utilization distribution of black-necked cranes in Huize National Nature Reserve (HNNR) in northeastern Yunnan on the Yunnan–Guizhou Plateau, China. In HNNR, common cranes (*G. grus*, Linnaeus, 1758), a species of least concern found within the family Gruidae (*BLI, 2018*), are also recorded. Common cranes are widely distributed across Eurasia with an estimated global population of *c.* 491,000–503,000 individuals (*BLI, 2018*). Based on their morphological similarity, interspecies competition between black-necked and common cranes likely exists. The competition exclusive principle predicts that at least one dimension of niche segregation is required for sympatric congeners (*Schoener, 1974*; *Holt & Lawton, 1994*; *Bagchi, Goyal & Sankar, 2003*). Thus, compared with former studies in areas where common cranes do not occur, we questioned whether the co-occurrence of common cranes impacts the habitat use and preference of black-necked cranes, particularly given the disparity in population numbers in our study area (common crane *c.* 350 individuals vs. black-necked crane *c.* 100 individuals) (*Kong, 2012*). We hypothesized that the dominant population would maintain the same habitat use and preference patterns observed in previous studies where only one species was distributed, with the disadvantaged population shifting their habitat use and preference.

At the same time, cranes are inclined to use habitats near their communal roosting sites to reduce energy expenditure (*Alonso & Alonso, 1992*; *Kong et al., 2011*). Thus, superior habitats near the roosting site may be occupied by the advantaged population, resulting in the spatial separation of the two species. The spatial use patterns (i.e., utilization distributions) of black-necked and common cranes were therefore considered in the present research.

We studied the wintering habitat selection (utilization and preference) of sympatric black-necked and common cranes in HNNR. We also compared our results with other studies in which only one species was distributed to determine if interspecies competition occurred. We then calculated the utilization distribution and distance to the roosting site to help clarify the spatial partitioning and determine areas of high priority for these species.

## MATERIALS AND METHODS

### Study area

This study was conducted between November 2010 and March 2011, covering the whole wintering period of both crane species. We conducted surveys in HNNR in northeastern Yunnan on the Yunnan–Guizhou Plateau (Fig. 1). The elevation of HNNR ranges from 2,470 to 3,092 m above sea level (*Qiou, 2012*). The reserve is divided into two discrete regions (Daqiao and Zhehai), which are located approximately 30 km apart. This study was conducted in the Daqiao region, which covers an area of 9,076.28 ha (N26°38′00″–26°44′24″, E103°12′06″–103°22′02″) (Fig. 1). The mean annual temperature

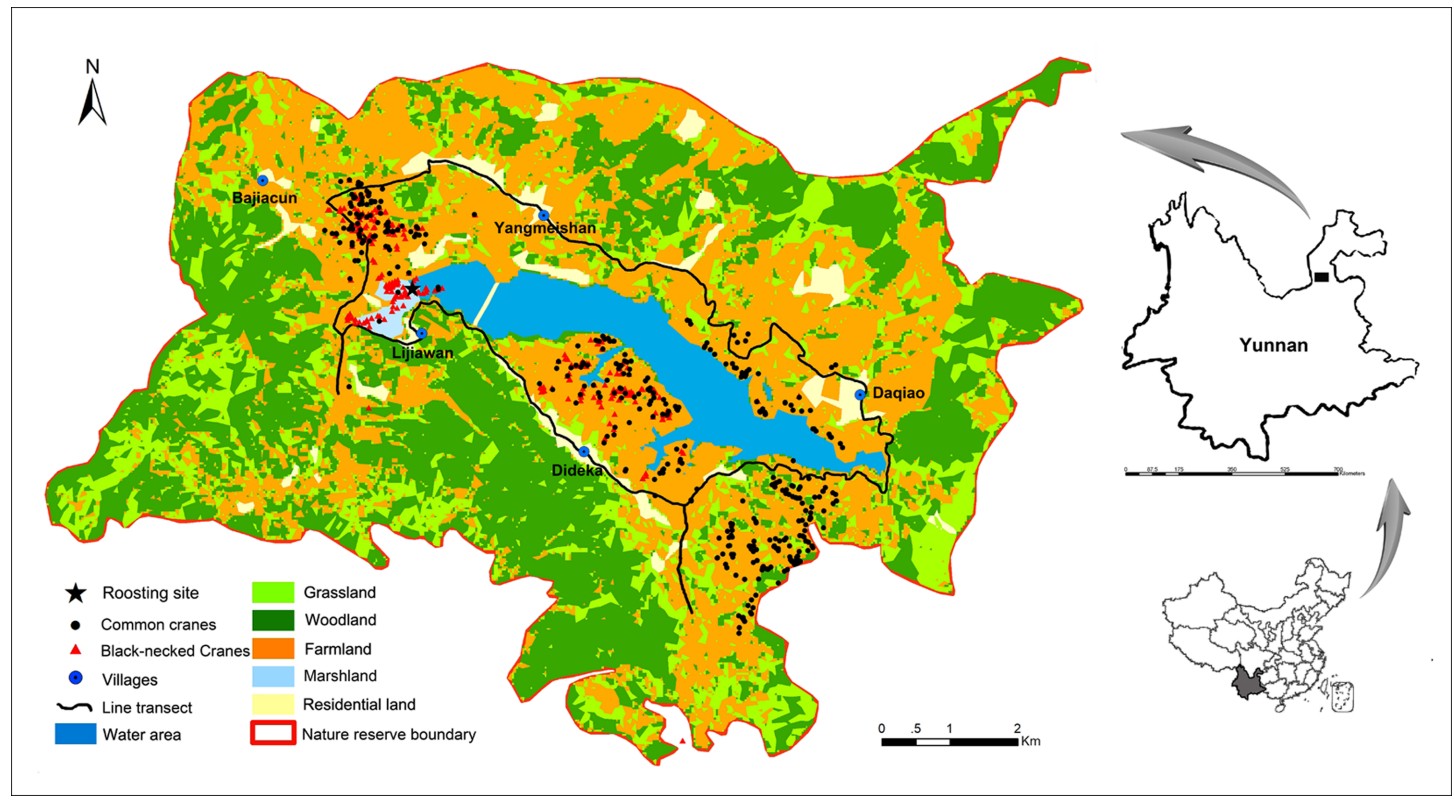

**Figure 1 Habitat use and spatial distribution of black-necked and common cranes in the Huize National Nature Reserve, northeastern Yunnan, China.**

at Daqiao is 9.6 °C, and the area experiences 40 days of snowfall, 50 days of snow on the ground, and 45 days of freeze-up annually (*Qiou, 2012*).

The Daqiao region can be further classified into areas of differing land use, including the Yuejin Reservoir (4,70.50 ha), marshland (149.36 ha), farmland (3,966.53 ha), grassland (178.19 ha), residential land (302.11 ha), and woodland (4,009.58 ha) (Fig. 1). The Yuejin Reservoir supplies a shallow water habitat for the roosting and foraging of wading birds. The surrounding marshland, farmland, and grassland also serve as foraging habitats for the crane species, whereas woodland is considered unsuitable habitat (*Kong et al., 2011*). As a typical anthropogenically impacted habitat, farmland experiences more intense human activity during the harvesting (October and November) and planting seasons (February and March).

Huize National Nature Reserve was first established in 1990 as a county-level reserve and upgraded to a national-level reserve in 2006 to protect wintering waterfowl and their habitats (*Qiou, 2012*). There are currently 100 black-necked cranes, 350 common cranes, and >3,000 individuals encompassing 63 other waterfowl species, including bar-headed goose (*Anser indicus*), ruddy shelduck (*Tadorna ferruginea*), grey heron (*Ardea cinerea*), recorded in the study area (*Qiou, 2012*). Both crane species are known as flagship species of the plateau wetland ecosystem (*Yang & Zhang, 2014*). The reserve also experiences intensive human disturbance due to the 12,250 people residing in the study area.

## Field surveys

Wintering cranes are gregarious and share communal roosting sites at night. They usually depart their roosting sites during the morning (06:30–08:00) to forage and return at night (18:00–20:00) (*Kong et al., 2008*). In the present study, we applied line transect surveys to record bird distributions and habitat use during their feeding times on clear days (no rain, snow, or fog) between 08:00 and 19:00 (*Krebs, 1998*; *Kong et al., 2011*). The line transects covered 30.2 km and were, on average, fully checked within 3 days (ranging from 2 to 5 days) by walking. Each transect started from the protection station in Yangmeishan village (Fig. 1). The end point along the line transect from the previous day was used as the start point on the following day. The continuous 3-day sampling was considered a complete survey, with 12 surveys accomplished in total. We switched the direction of travel for the next complete survey. The 12 surveys were distributed over the five months of the wintering period (including one in November, three in December, two in January, four in February, and two in March). We recorded all crane flocks within the field of vision of 10 × 42 binoculars, and the width of the transects varied with visibility. We visually classified a multi-temporal Landsat TM 5 satellite image (captured on 14 March 2011) into six different land-use types encompassing farmland, grassland, marshland, woodland, water area and residential land. Then, we conducted the viewshed analysis to get the land use data alongside the line transect using Global Digital Elevation Mode version 2 in ArcGIS 9.3 (ESRI, Redlands, CA, USA). In total, the transect area covered 5,001.62 ha and included 2,216.65 ha of farmland, 760.39 ha of grassland, 38.34 ha of marshland, 2,216.65 ha of woodland, 321.43 ha of water area, and 236.88 ha of residential land. We defined flocks as being discrete if they were more than 500 m apart. Each flock was considered as a sample unit and one GPS point was recorded (*Thomas & Taylor, 1990*). All crane flocks were marked in Google Earth with an Android device. For each flock of cranes, we also recorded roosting site distance, which was defined as the distance from the location of each flock to the communal roosting site (N26°42′05.6″, E103°16′00.6″) and was calculated in ArcGIS 9.3. Field studies were conducted under the permission from the Administrative Bureau, National Nature Reserve of Black-necked Cranes in Huize.

We only considered farmland, marshland, and grassland as available foraging habitats for cranes, as indicated in former studies (*Kong et al., 2011*). Farmland included plowed and unplowed land used for crops such as *Solanum tuberosum*, *Brassica campestris*, and *Zea mays*. Marshland was located near reservoirs and was covered with shallow water (≤50 cm) throughout winter. Dominant vegetation in marshland included *Ranunculus japonicus*, *Juncus effusus*, and *Poa annua*, whereas the dominant vegetation in grassland included meadows of *Leontopodium andersonii*, *Primula malacoides*, and *Trifolium repens*.

## Habitat use and preference

Habitat use was calculated by the number of crane flocks occurring in each habitat type as a percentage of all crane flocks observed. We used the relative habitat use indicator of Ivlev's electivity index ($s$) to evaluate habitat preference for each sample and habitat (*Ivlev, 1961*; *Wood & Stillman, 2014*). The electivity index was calculated as $s = (a - b)/(a + b)$, where $a$ is the percentage of flocks using a given habitat and $b$ is

the habitat area as a percentage of the total available habitat area (*Jacobs, 1974*). For each habitat, we obtained an electivity value ranging from −1.0 (never used) to +1.0 (exclusively used), with 0.0 representing habitat used in proportion to its availability (*Ivlev, 1961*). Thus, positive and negative electivity values represent habitat preference and avoidance, respectively. Seasonal habitat preference was also considered for each species during the wintering period from November to the following March.

## Utilization distribution

Utilization distribution provides a convenient global representation of spatial use patterns by defining the relative frequency of occurrence of animals (*Benhamou & Riotte-Lambert, 2012*). We calculated utilization distributions using the nonparametric kernel local convex hull (LoCoH) method to assess spatial use by the studied cranes (*Getz & Wilmers, 2004*; *Getz et al., 2007*). This method is more appropriate than parametric kernel methods for constructing utilization distributions and can capture hard boundaries (e.g., rivers and cliff edges) and process large sample sizes (*Getz et al., 2007*). This method is also very powerful in processing aggregated and clustered data (*Getz & Wilmers, 2004*) at the population level (*Liu et al., 2010*). Thus, we constructed kernels with the fixed radius local convex hull (*r*-LoCoH) method (available at http://locoh.cnr.berkeley.edu) using flock location data within a fixed 500 m radius, which was sufficient to distinguish flocks of the two-crane species. The obtained shapefiles were imported into ArcGIS 9.3 to construct utilization distribution maps. We considered 90% instead of 100% isopleths as the overall crane distribution range by omitting outlying points representing exploratory animal movement rather than that necessary for survival. The 90% utilization distribution isopleths can faithfully reflect actual spatial distribution patterns of animals (*Börger et al., 2006*). For protection management, 70% and 50% isopleths of utilization distribution are usually recognized as the ordinary and kernel distribution range of wildlife. Thus, we considered 90%, 70%, and 50% utilization distribution isopleths in the current study to determine areas of high conservation priority.

## Statistical analysis

We did not assess seasonal habitat preference differences because of the small sample size each month. Considering the lack of independence of the 12 surveys of the same study area, pseudoreplication may occur (*Hurlbert, 1984*). So, we implemented a general linear model to compare the differences in distance to roosting site for the two species, with the survey order as random effect; and the sum of squares type III was selected in the model. Statistical analysis was completed with IBM SPSS Statistics 19.0. We regarded differences between two variables as statistically significant and highly significant when the two-sided *p*-values were <0.05 and <0.01, respectively. Averages were presented as mean ± SD.

# RESULTS

## Habitat use and preference

In total, we observed 285 black-necked crane flocks and 387 common crane flocks. In winter, both species showed a similarly high proportion of farmland habitat use, but

**Table 1 Habitat availability (%), use (%) and preference (s) of black-necked and common cranes in Huize National Nature Reserve, northeastern Yunnan, China.**

|  |  | Habitat type | | | Total |
|---|---|---|---|---|---|
|  |  | Farmland | Marshland | Grassland |  |
| Black-necked cranes | Area (ha) | 2,216.7 | 38.3 | 760.4 | 3,015.4 |
|  | Habitat availability (%) | 73.5 | 1.3 | 25.2 | 100.0 |
|  | No. of flocks | 265.0 | 19.0 | 1.0 | 285.0 |
|  | Habitat use (%) | 93.0 | 6.7 | 0.4 | 100.0 |
|  | s (mean ± SD, n = 12) | 0.11 ± 0.01 | 0.02 ± 0.26 | −0.97 ± 0.03 | – |
| Common cranes | No. of flocks | 365.0 | 2.0 | 19.0 | 386.0 |
|  | Habitat use (%) | 94.6 | 0.5 | 4.9 | 100.0 |
|  | s (mean ± SD, n = 12) | 0.12 ± 0.01 | −0.73 ± 0.19 | −0.76 ± 0.08 | – |

Notes:

Habitat preference was evaluated using Ivlev's electivity index as $s = (a - b)/(a + b)$, where $a$ is the percentage of flocks using a given habitat and $b$ is the habitat area as a percentage of total available habitat area. Positive and negative electivity values indicate habitat preference and avoidance, respectively.

different habitat use patterns for marshland and grassland (Table 1). We only recorded one common crane flock of four individuals in a marginal woodland area, and no black-necked cranes at all. Thus, woodland was considered as an unavailable or unexploited habitat and was excluded from the following calculations.

During winter, both black-necked and common cranes preferred farmland (positive selection) and avoided grassland (negative selection) (Table 1). In contrast, common cranes avoided marshland, whereas black-necked cranes showed a seasonal though changing preference for this land type. Specifically, black-necked cranes avoided marshland in the first three months of the wintering period (November, December, and January), but showed a preference for it in spring (February and March), even higher than that for farmland (Fig. 2).

## Utilization distribution

The black-necked and common cranes were distributed in relatively separate areas (Fig. 3). For the black-necked cranes, 58.60%, 40.35%, and 1.05% of flocks were distributed in the Baijiacun–Lijiawan, Maanshan, and Dideka-Daqiao areas, respectively; whereas, for the common cranes, 23.58%, 22.28%, and 54.15% of flocks were distributed in Baijiacun–Lijiawan, Maanshan, and Dideka-Daqiao areas, respectively.

We found that the overall (90% isopleths), ordinary (70% isopleths), and kernel (50% isopleths) utilization distributions of black-necked cranes were smaller than those of common cranes (Table 2; Fig. 3). Accordingly, compared with the common cranes (distance to roosting site: 4.38 ± 0.11 km, $n = 386$), the black-necked cranes were detected in areas at significantly shorter distances to the roosting site (1.89 ± 0.08 km, $n = 285$, $f = 66.49$, $p < 0.01$). We also found a significant interactive effect of survey order and species on the distance to roosting site ($f = 3.37$, $p < 0.01$).
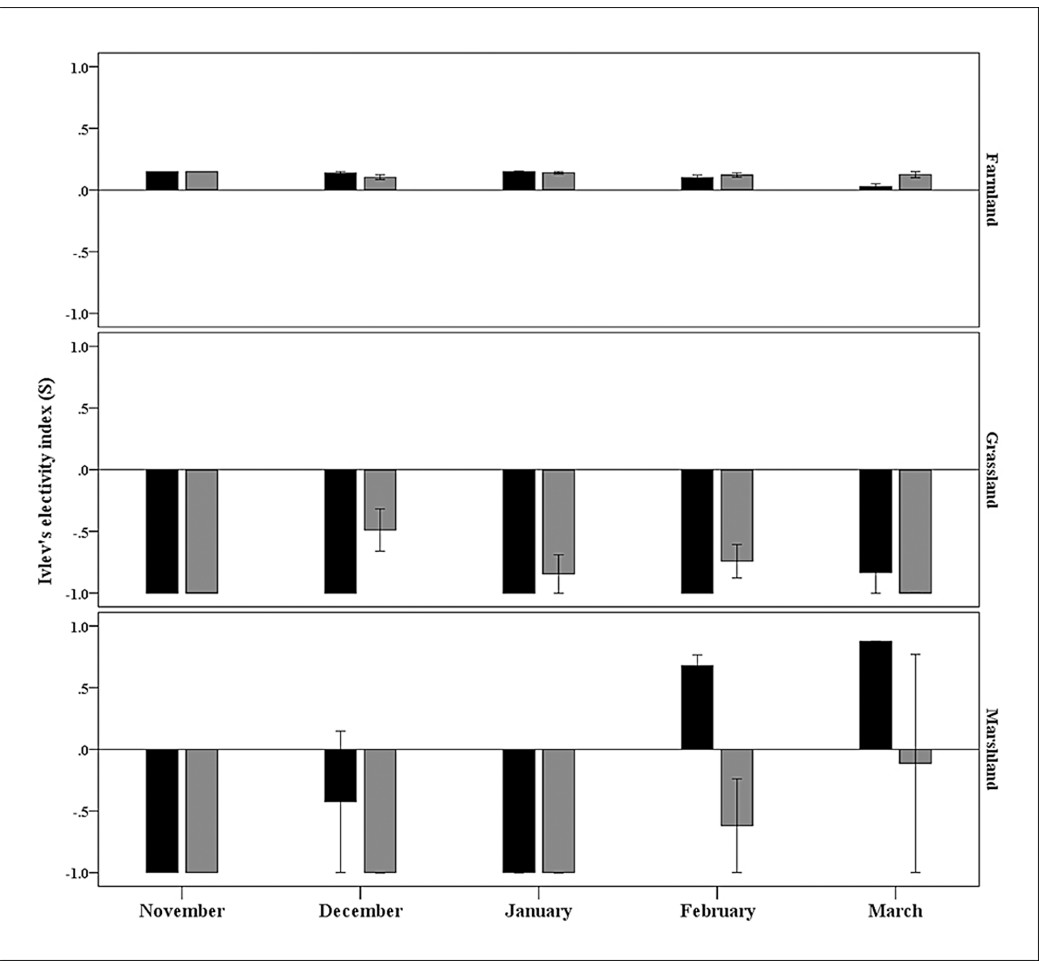

**Figure 2** **Seasonal habitat preferences of black-necked (black bars) and common cranes (gray bars) in the Huize National Nature Reserve, northeastern Yunnan, China.** Positive and negative electivity values indicate habitat preference and avoidance, respectively.

## DISCUSSION

Black-necked and common cranes are recognized as flagship wetland species on the Yunnan–Guizhou Plateau (*Yang & Zhang, 2014*). Due to their close phylogenetic relationship and similar morphologies, these birds boast similar wintering ecologies. We found that the both species exhibited high dependency on anthropogenic farmland habitat during winter, which was not unexpected given that farmland occupies the highest proportion of available habitat (73.5%) in the research area. In accordance with our study, wintering black-necked cranes have been reported to forage frequently in farmlands in Dashanbao Nature Reserve (*Kong et al., 2011*) and Yongshan County (*Lu & Yang, 2014*) on the Yunnan–Guizhou Plateau, and in the Lhasa River Valley of Tibet on the Qinghai–Tibetan Plateau (*Tsamchu & Bishop, 2005*). The higher proportion of farmland habitat use by black-necked cranes is likely the result of higher food availability in farmland than in other habitats (*Li et al., 2009*). For example, remnant crops, such as potatoes (*S. tuberosum*), oats (*Avena sativa*), buckwheat (*Fagopyrum tataricum*), and

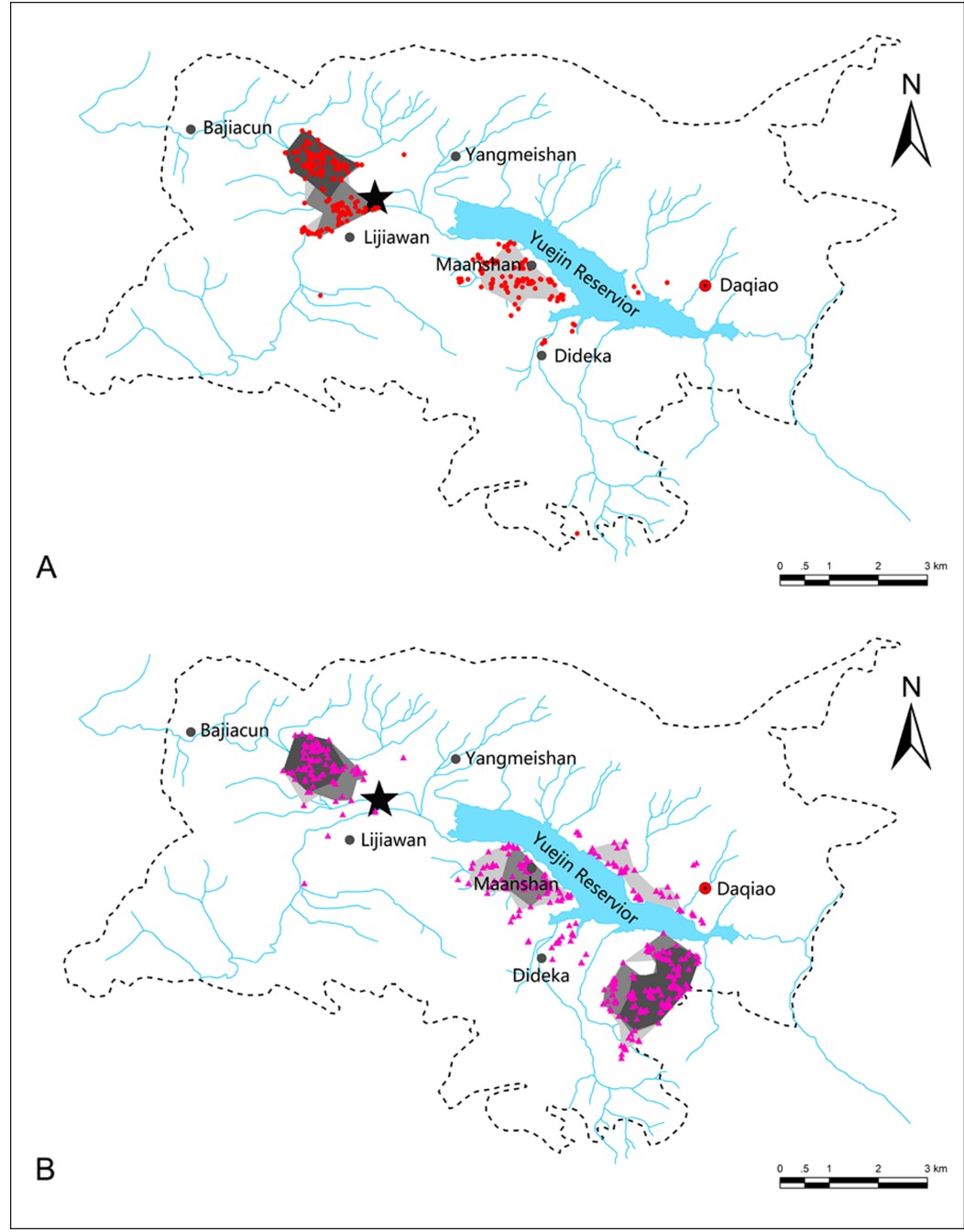

**Figure 3 Utilization distributions of black-necked (A) and common cranes (B) in the Huize National Nature Reserve, northeastern Yunnan, China.** Dark gray, gray, and light gray areas represent the 50%, 70%, and 90% isopleths of utilization distribution of each species. The black star indicates the communal roosting site.                                         

corn (*Z. mays*), are reported to supply over 80% of wintering food for black-necked cranes (*Dong et al., 2016*).

Farmland and marshland rather than grassland were favored by black-necked cranes in HNNR, the same as reported in the Dashanbao Nature Reserve (*Kong et al., 2011*).

**Table 2 Utilization distributions (UDs) of black-necked and common cranes in Huize National Nature Reserve, northeastern Yunnan, China.**

|  | 90% Isopleths of UD (ha) | 70% Isopleths of UD (ha) | 50% Isopleths of UD (ha) |
|---|---|---|---|
| Black-necked cranes | 283.84 | 168.58 | 92.89 |
| Common cranes | 558.73 | 380.46 | 224.81 |

Note:
The nonparametric kernel local convex hull (LoCoH) method was used in the calculation of utilization distribution to assess spatial use by the studied cranes (*Getz & Wilmers, 2004*; *Getz et al., 2007*).

**Table 3 Habitat availability and composition in three national nature reserves (Huize, Dashanbao, and Caohai Reserves) on the Yunnan–Guizhou Plateau.**

|  | Habitat availability % | | | |
|---|---|---|---|---|
|  | Farmland | Marshland | Grassland | Sedge meadow |
| Huize reserve, northeastern Yunnan | 73.5 | 1.3 | 25.2 | – |
| Dashanbao reserve, northeastern Yunnan | 27.5 | 10.5 | 62 | – |
| Caohai reserve, Guizhou | 54.3 | 12.6 | 4.9 | 28.1 |

However, the black-necked cranes in this study preferred farmland to marshland, whereas the cranes in Dashanbao preferred marshland over other habitats (*Kong et al., 2011*) and the cranes in Caohai Reserve preferred sedge meadow (*Li, 1999*). These distinctions are probably due to the habitat availability differences among the different wintering sites (Table 3). The very high proportion of farmland habitat in HNNR resulted in its intense use and preference over other habitats. However, black-necked cranes also showed a very strong preference for marshland in February and March (Fig. 2). This is probably due to the increase in behaviors such as preening, singing, and dancing in spring (*Kong et al., 2008*), which are performed to establish or enhance pair bonding for the upcoming breeding season, with marshland reported to provide optimal areas for such social behaviors (*Kong et al., 2011*). In addition, intense human disturbance from spring plowing in February and March could force cranes from farmland and thereby influence their preference for marshland.

Both black-necked and common cranes avoided grassland in the current study, which was possibly due to low food availability (*Li et al., 2009*). Most common crane flocks were detected in farmland, in agreement with other studies from Asia and Europe (*Avilés, 2003*; *Zhan et al., 2007*), but avoided marshland and grassland, in disagreement with earlier studies where farmland and marshland were favored habitats when black-necked cranes were absent (e.g., Beijing Yeyahu Wetlands (*Zhan et al., 2007*); Spain (*Avilés, 2003*)). Thus, we determined that common cranes preferred farmland regardless of the presence of black-necked cranes. Our results also verified that habitat preference established by innate and learned behavioral decisions reflected the biological characteristics of the animals (*Hall, Krausman & Morrison, 1997*). However, the low proportion of available marshland (1.3%) in our study area may have influenced the extremely low use of this habitat (0.5%) by common cranes. When wintering with black-necked cranes in sympatric areas, common cranes avoided the less available

marshland. We inferred that this may be caused by the presence of black-necked cranes, whose larger body size provides them with an advantage when competing for resources in favored habitats (*Smith & Brown, 1986*). Thus, the differences in habitat preference between this study and others may be explained, at least in part, by interspecies competition.

Coexistence can occur for similar species when niche divergence is present (*Schoener, 1974*; *Dufour et al., 2015*; *Xia et al., 2015*). However, we found that the cranes avoided interspecies competition by moderate divergence of habitat preference, as mentioned above. We also found significant segregation between the two species in spatial distribution. Both crane species avoided foraging together in winter by dispersing in different areas. Nearly all black-necked cranes (99.0%) were distributed in the Baijiacun–Lijiawan and Maanshan areas, whereas over half of the common cranes (54.15%) were distributed in the Dideka–Daqiao area (Fig. 3). Previous empirical observations have indicated that black-necked and common cranes share roosts but compete for foraging sites when wintering in sympatry (*Li & Li, 2005*), and have often been detected foraging at different sites in the Napahai Wetlands on the Yunnan–Guizhou Plateau (*Yang, Huang & Guan, 1992*). Our results showed that black-necked cranes concentrated their foraging in the low-lying areas near the common roosting site, whereas the common cranes occupied larger areas on hill sides. We also found that common cranes frequently selected habitats up to 4.38 km from their roosting sites. Earlier studies revealed that foraging near roosting sites is a strategy used by cranes to reduce energy expenditure (*Alonso & Alonso, 1992*), and only a dominant species can occupy the optimal habitat, e.g., close to the roosting site or with sufficient food (*Kong et al., 2011*). We occasionally observed the larger black-necked cranes repelling the smaller common cranes from their foraging habitat. Observations in the Caohai Nature Reserve on the Yunnan–Guizhou Plateau also suggest that black-necked cranes mostly forage in places near their roosting site, whereas smaller common cranes forage in peripheral areas 10–20 km away (*Yang, Huang & Guan, 1992*). At the same time, with larger populations, common cranes may need to occupy more expansive areas than black-necked cranes.

Taking into consideration our results and those of earlier habitat studies, we inferred that cranes use different habitats in different ways (*Kong et al., 2011*; *Dong et al., 2016*). Marshland may be recognized as the optimal foraging habitat for cranes because of considerable food resources (including underground tubers and insect larvae), soft ground surfaces for digging, and difficult access for humans (*Li et al., 2009*; *Kong et al., 2011*). Marshland was found to be a vital area for black-necked crane socializing (*Kong et al., 2011*). Although farmland contains the largest amount of underground tubers and considerable insect populations, this habitat is considered suboptimal due to higher human disturbance (*Li et al., 2009*; *Kong et al., 2011*). Despite this, farmland is highly utilized by most cranes (especially for black-necked cranes) across the Yunnan–Guizhou to Qinghai–Tibetan plateaus (*Tsamchu & Bishop, 2005*; *Kong et al., 2011*; *Lu & Yang, 2014*), and can be regarded as vital foraging habitat during winter. With scarce food resources and hard ground surfaces, grassland represents the poorest crane habitat (*Li et al., 2009*; *Kong et al., 2011*).

Although this study was carried out at only one site, our findings may shed light on other mountain areas with similar landscapes. This research should also provide a valuable resource for habitat conservation and protected area management. Our results indicated that effective and sustainable conservation measures, such as maintaining farmland, restoring wetlands, and prohibiting humans and livestock from entering core crane areas, could benefit wintering crane species. We believe that the conservation of flagship crane species could also enhance conservation efforts for other waterbirds in the wetland system.

## CONCLUSIONS

As two closely related species, black-necked and common cranes showed high similarity in habitat use. However, they were inclined to utilize habitats in different areas. Black-necked cranes maintained a central area near the common roosting site, whereas common cranes inhabited larger areas and at further distances from the roosting site. We argue that spatial separation could mitigate interspecies competition and facilitate coexistence. We recommend protection of the farmlands utilized by cranes and the restoration of additional wetland areas.

## ACKNOWLEDGEMENTS

We greatly appreciate the field assistance provided by all staff from the Huize National Nature Reserve. We are also grateful to Beverly Pfister, Elena Smirenski, and Fengshan Li for their invaluable editing of the manuscript and comments. Many thanks to the three anonymous reviewers and academic editor from PeerJ for their valuable comments which improved the manuscript a lot.

### Funding

This work was supported by National Natural Science Foundation of China (31201725) and the Applicable Basic Research Project of Yunnan Province (2012FB186). The funders had no role in study design, data collection and analysis, decision to publish, or preparation of the manuscript.

### Grant Disclosures

The following grant information was disclosed by the authors:
National Natural Science Foundation of China: 31201725.
Applicable Basic Research Project of Yunnan Province: 2012FB186.

### Competing Interests

The authors declare that they have no competing interests.

### Author Contributions

- Dejun Kong conceived and designed the experiments, performed the experiments, analyzed the data, prepared figures and/or tables, authored or reviewed drafts of the paper, approved the final draft.
- Weixiong Luo performed the experiments, analyzed the data, prepared figures and/or tables, approved the final draft.
- Qiang Liu analyzed the data, contributed reagents/materials/analysis tools, prepared figures and/or tables.
- Zhuoqing Li contributed reagents/materials/analysis tools, prepared figures and/or tables.
- Guoyue Huan performed the experiments, approved field study permission.
- Jianjun Zhang performed the experiments.
- Xiaojun Yang conceived and designed the experiments, approved the final draft.

## Field Study Permissions

The following information was supplied relating to field study approvals (i.e., approving body and any reference numbers):

Field studies were conducted under the permission from the Administrative Bureau, National Nature Reserve of Black-necked Cranes in Huize.

## Data Availability

The raw data are provided in a Supplemental File.

## Supplemental Information

Supplemental information for this article can be found online at http://dx.doi.org/10.7717/peerj.5105#supplemental-information.

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
