# Peer review of "Habitat use, preference, and utilization distribution of two crane species (Genus: Grus) in Huize National Nature Reserve, Yunnan–Guizhou Plateau, China"

_PeerJ, doi:10.7717/peerj.5105_

## Round 0.1 · original submission · Major Revisions

Reviewers 1 and 2 recommended publication following a relatively small number of changes. Although neither reviewer commented on the statistical analysis, I was concerned about several aspects and therefore sought the advice of a third reviewer who had carried out a somewhat similar study. This reviewer agreed that your study would be a useful contribution to the scientific literature, but confirmed my worries about the analysis. He also detected a number of other substantial problems. My decision is that the manuscript requires major revisions. Because some of these revisions involve the statistical analysis and other methods, the manuscript may have to be sent out for review a second time. You should consult a statistical advisor to be sure that your corrections to the statistics are appropriate.

In addition to the reviews, I have provided comments of my own and an annotated pdf. Reviewers 1 and 2 provided many comments on pdf versions of the manuscript. Some of the most important comments of Reviewer 2 were stated in somewhat in an informal style that might not be clear to a non-native English speaker; I have therefore explained these comments below. In my annotated manuscript, I used orange highlights to indicate sections of concern and inserted comments to suggest alternative wording or explain the problem. In the case of repeated mistakes, I sometimes highlighted them without re-stating the problem. There are many more problems with language than I have time to correct. In some cases, I highlighted errors without explanation and in other cases did not have time to note all problems. After making all the changes suggested by the reviewers and editor and making your revisions, you should obtain a careful, detailed review of the writing by a native English speaker.

Editor's Comments

The Introduction requires elaboration of several aspects:
• The knowledge gap that your study is intended to fill is not clearly identified. This requires a review of previous relevant literature, including your own previous research on habitat use of cranes. This would not be as detailed as the Discussion but is needed so that the reader knows what was previously found.
• The Introduction (not the Methods) is where the concepts of habitat use, preference and utilization distribution and their relevance should be introduced.
• The objectives do not explicitly include the study of flock size, elevation and distance from roost. The Introduction must help the reader see the importance of these measurements and the objectives should include them.

All reviewers mentioned gaps in the methods.
• As the other reviewers noted, the width of your transect is not stated. If this varied with visibility, you need to give the reader a way to understand how you carried it out and what the total area of the transect was.
• You need to explain the source of data on land cover and how the area of each habitat type was calculated. You must indicate whether this measure applied to the general area of the reserve or only to the area covered by the transect. This is a critical point for the validity of your preference measure.
• For preference, you used the compositional analysis method of Aebischer et al. 1993 which is appropriate for statistical analysis of multiple samples. However, you obtained only a single value, so a simple ratio of use to availability would provide exactly the same rank order of preferences. As suggested by reviewer 3, you may want to consider switching to Ivlev's index and take into account the variation inherent in your repeated sampling of the same transect.

Style and format:
• Your manuscript is justified on both the right and left margins. This makes it harder to read, which is why PeerJ requests left justification only.
• Also, the manuscript seems to be single spaced. Double or 1.5 spacing would make it easier to edit.
• For ease of reader comprehension, keep the species in the same order throughout the entire manuscript, starting with the Abstract.
• Abbreviations should be defined the first time a word or expression is used and then used consistently without the word or expression in the rest of the manuscript. Readers do not like to remember multiple abbreviations, so only use a few abbreviations for the most frequently used terms. I think it is better not to abbreviate your study species names.
• Figures 2 and 3 are missing captions.
• You should avoid excessive overlap in presenting data. It appears that Table 2 and Fig. 2 completely overlap. You only need one of them. Reviewer 3 suggests that the figure would be best. You could add the sample sizes by placing numbers above each box plot. The caption needs to define the box plot type.

Editor's clarifications of manuscript annotations by Reviewer 2 (Daniel Collins)

CD1. Although your statement is correct, readers may miss the distinction between utilization and preference. I suggest the following wording:
The line transect method indicated that the anthropogenic habitat of farmland was utilized the most by these two species (>90% of flocks observed for both). However, an analysis based on the ratio of use to availability indicated that Black-necked Cranes preferred marsh to farmland and grassland while Eurasian Cranes preferred grassland.
CD2. I agree that the Introduction should be expanded to explain why habitat use and spatial distribution are important.
CD3. I think that the reviewer means you should use the present tense when describing the current status of your study species, and I agree.
CD4. I think all you need to say is that 'Both species are migratory.' The rest of the paragraph provides sufficient explanation of the differences.
CD5. Provide population size for Eurasian Cranes, if available.
CD6. I believe that it is acceptable to use an acronym such as HNNR to start a sentence in PeerJ. (This is an acceptable abbreviation.)
CD7. The statement indicating the number of cranes and other species is appropriate for the Study Area section and does not need to be removed. However, there should be a reference for the source of this information.
CD8-10. I suggest the following revision:
Three separate areas comprising the villages of Yangmeishan-Bajiacun-Lijiawan (YBL), Maanshan (MAS) and Daqiao-Dideka (DD) were included along the transect (Fig.1). The transect was 16.6 km long and required two days to complete. On the first day, we started at the protected station in Yangmeishan village. 
CD11. If you change 'population' to 'local population', your meaning will be clear.
CD12. Yes, information is needed regarding whether the GPS location was the observer's position or the crane's position. If is was the crane flock position, explain how you determined the location without disturbing the flock.
CD13. The abbreviation is acceptable.
CD14. The only statistical analysis you did involved the data on altitude, distance and flock size. You need to be clear in this section what questions the analysis was directed toward. Presently, it is unclear and incomplete. However, you do not need to make it shorter as implied by the reviewer's use of the word 'succinct'. (There will be additional statistical methods if you follow suggestions of reviewer 3.)
CD15. I disagree with the reviewer here. I think you defined use with sufficient clarity and that your preference measure reduces the need for a chi-square test on use. Your discussion does address the issue of farmland that is heavily used even though it is less preferred.
CD16-20. The reviewer is correct; the first two sentences should be removed. Start with
'We found wintering crane species . . . '. This should be a new paragraph so that you have one paragraph for each species. Try to organize the information better so that you put all the habitat comparisons together and then discuss how diet could explain the patterns. You do not need to repeat the concept of habitat use from the Introduction in this paragraph (or the next one).
CD21. You should present any other available evidence to support the argument that these two species (or similar ones) compete and one may displace the other from foraging sites. If this is the first evidence for possible competition between these species, you should state that explicitly.
CD22-23. The reviewer is suggesting that you investigate the literature a bit further to determine additional evidence related to the relationship between foraging behavior and habitat use and preference (probably on other species).
CD24. If your analysis provides any information about human or livestock impacts on habitat use, you should discuss it earlier. If you don't have any evidence, it is not logical to make a recommendation. Rather, the recommendation should be that human and livestock impacts should be investigated in the future.
CD25. Because the Discussion is quite short, I don't think you need a longer Conclusion. The Conclusion should be just a brief reminder of what you have already written.

Reviewer 1 ·

Basic reporting

The manuscript by Kong et al is clear and well written. The Introduction addresses properly the topic of the Study, while Study Area, Methods and Results are well structured and unambiguous. Figures and Tables are clear.
In the Conclusions section, the final sentence (lines 295-296) is rather vague; I suggest to add some details or examples of possible management strategies useful in the context studied by the Authors.
A few minor notes:
1. In the pdf version of the ms I highlighted some spelling mistakes;
2. Line 108. Please add at least one source for the transect method;
3. Line 120. An important detail which is missing is the distance from the transects (up to 500 m? unlimited? other?) within which birds were counted;
4. Thomas & Taylor 1990 is not included in the References list;
5. Line 172. “Luca” is the name of the first author, not the surname. See References list for the correct quotation.

In conclusion, I recommend the manuscript for publication.

Experimental design

No comment

Validity of the findings

No comment

Additional comments

No comment

Annotated reviews are not available for download in order to protect the identity of reviewers who chose to remain anonymous.

·

Basic reporting

In general this paper is written in a way that is understandable but has a couple areas of improvement and expansion.

Experimental design

The methods section of this paper need improvement and expansion which have been capture in the attached track changes.

Validity of the findings

Results need to be expanded and certain areas of the discussion need work.

Additional comments

In general the paper is written well but could be improved with some expansion and recasting. In the end the paper will help inform conservation within the area but the story you have laid out lacks some details to holistically tell that story and identify those conservation needs.

Reviewer 3 ·

Basic reporting

In general the manuscript is well presented, with a professional article structure which is set out logically. However, the rationale for carrying out the study could be developed more clearly. In the current version of the manuscript the knowledge gap is not identified clearly. Without this rationale, it will be hard for the reader to understand why the study has been carried out in the way it has. The authors state on on lines 71-75 that information on habitat use and distribution are needed to "put forward more rational and effective habitat management", but no information is presented on current habitat management and why this is ineffective, or why information on habitat use and preferences will help to address the knowledge gap. Some of the text currently in the methods section (e.g. lines 138-143 and 156-158) should be moved to the introduction and used as part of an improved rationale for the study.

The manuscript contained some spelling and grammatical errors, and so I recommend that the authors consult a native English speaker to correct these. I have listed a few examples below, but there are many others:
- line 49: "migrants" not "migrators".
-line 72: "measures" not "measurements".
- line 249: "segregation" not "segregating".

There were a number of instances where the citation of additional literature would add useful additional detail or clarification to the manuscript:
- lines 46-48: Is there a reference to the Biodiversity Red List of China that can be cited?
- lines 71-72: You do not say why information on habitat use and preferences will help to "put forward more rational and effective habitat management". There are previous studies of waterbird habitat use and preferences that you could cite to help develop your argument, e.g. Wood & Stillman (2014) Do birds of a feather flock together? Comparing habitat preferences of piscivorous waterbirds in a lowland river catchment. Hydrobiologia (2014) 738:87–95. DOI 10.1007/s10750-014-1921-6
- lines 150-154: If you decide to use the simpler method of estimating habitat preferences, this is the original reference for Ivlev's electivity index: Ivlev, V. S. (1961). Experimental ecology of the feeding of fishes. Yale University Press, Connecticut.
- lines 174-179: To aid with the statistical analyses, I have recommended reading and citing the following paper: Zuur, A. F., E. N. Ieno & C. S. Elphick, 2010. A protocol for data exploration to avoid common statistical problems. Methods in Ecology and Evolution 1: 3–14.
- lines 174-179: If you choose to retain modified versions of the statistical analyses that account for the sources of pseudoreplication, then I recommend citing the following paper in your explanation: Hurlbert, S. H. (1984). Pseudoreplication and the design of ecological field experiments. Ecological Monographs, 54, 187-211.

Experimental design

In my view the manuscript certainly represents original primary research that falls well within the aims and scope of PeerJ.

As mentioned above I felt that the rationale for carrying out the study could be developed more clearly. I also had more substantial concerns regarding the data collection and calculations:

1) Field surveys
The description of the methods could be improved, as the current version of the manuscript omits many details which are required to evaluate the study. Whilst the length of the transect is given (line 112), the width of the transect is not. What area either side of the transect was surveyed? Were the 12 surveys spaced out equally across the study period?


2) Habitat preference calculations
There are many different available indices of preference, and so it is important to explain the choice. Why was the ranking matrix used rather than the simpler electivity index proposed by Ivlev (1961)? The Ivlev electivity index is particularly useful as it presents values on a scale of -1.0 (total avoidance of habitat type) to +1.0 (exclusive use of habitat type) which is widely understood and easy to interpret, and this is what I recommended that you use in your study. If you wish to use a more complex method, you should give compelling reasons for doing so. Furthermore, in calculating habitat preferences it is important that the area of habitat used to estimate habitat availability matches the area used to determine the habitat use of the birds. However, it is not clear from the methods reported here whether this was the case. Finally, reporting single values of habitat use and preference ignore the variance between the 12 surveys. Instead, mean values with 95% confidence intervals would be much more useful than the single values for habitat use and preference that are currently presented. These mean +/- CI habitat use and preference values would be more effectively displayed in figures rather than in tables, as figures would allow the differences between habitat types and species to be visualized more easily.

Validity of the findings

In general I felt that the discussion was well written, with appropriate references to the key findings and their context. As I have recommended a number of changes to the analyses, some of the conclusions may change as the manuscript is revised.

Statistical analyses
This section requires much additional detail in order to allow the reader to understand what was done and why. What parametric tests were used, as none are mentioned after line 174?
In lines 174-177 you state that you assessed whether your data conformed to a normal distribution. However, this is not the correct procedure. The parametric tests that you mention assume normality of the test residuals, not the raw data. Therefore, it is the residuals that need to be tested for normality. Many parametric tests make other assumptions that must be met, for example homoscedasticity. Were these assumptions also tested? I recommend the following paper on approaching statistical testing: Zuur, A. F., E. N. Ieno & C. S. Elphick, 2010. A protocol for data exploration to avoid common statistical problems. Methods in Ecology and Evolution 1: 3–14.
It was not clear to me what questions were being tested in the analyses mentioned here. I recommend revising this section to state the purpose of each test, including which dependent and independent variables were included.
The testing of the effects of (i) elevation, (ii) distance to roost, and (iii) flock size is problematic, as it was not properly introduced in the manuscript and there are issues of pseudoreplication (sensu Hurlbert 1984) that are not addressed. For example, the data combine 12 surveys of the same study area. Because the 12 surveys are of the same area (and presumably contain many of the same flocks) each survey cannot be considered to be independent. There is nothing reported in your manuscript that tells the reader how you addressed this problem. All of the data cannot simply be combined into a single analysis.If you wish to keep this section, then the analysis must be properly introduced and properly carried out. The sources of non-independence must be accounted for, perhaps via a repeat measures ANOVA or a mixed-effects model with survey data modelled as a random effect. The precise method that will be most appropriate will depend on how well the test assumptions are satisfied.
Finally, when conducting multiple tests and assessing significance via P values, you must correct your P values for multiple comparisons to maintain the level of alpha at 0.05. I recommend using Holm-Bonferoni adjustments to your P values; I calculated the adjusted P values the 7 values that you present in lines 189-194, and found that while all othe highly significant results remained <0.001, the values of 0.032 and 0.045 both became 0.064 (i.e. they became non-significant).

There were a number of other areas in which I felt that the manuscript could be improved:
The individuals within flocks are non-independent, and so basing your calculations of habitat use and preference on flocks seems reasonable. However, it would be valuable to be able to see how many individuals were using each habitat type, to provide the reader with useful information on the numerical importance of each habitat type. Perhaps this could be included in Table 1?
The percentages associated with the flock field utilization values are presented to too great a level of precision given the methods used; presenting these values to one decimal place would be sufficient. The same comment applies to the values reported in Tables 1, 2 and 3.

Additional comments

The manuscript contains many interesting ideas and I enjoyed reading it. You will see from my comments in the previous sections that I felt that there were a number of areas where the manuscript would be improved by revisions, in particular to the analytical methods. Whilst I know that such a large number of comments will always seem disheartening at first, I hope that I have explained adequately why I felt that each of the changes was necessary, and why I think that it will ultimately make your manuscript better. I hope that you will find my comments useful in improving your manuscript.

---

## Round 0.2 · accepted · Accept

The reviewer and I consider that you have done an excellent job of revising your manuscript. I appreciate your attention to the numerous comments. I believe that the manuscript is now suitable for publication. I have attached a pdf with some minor corrections in the text, tables, and figure captions. These changes can be made during the publication process.

# Reviewer 3 ·

Basic reporting

no comment

Experimental design

no comment

Validity of the findings

no comment

Additional comments

I have now read the revised version of the manuscript. This revised manuscript represents a great improvement on the previous version. In my view all of the points raised in the earlier reviews have been addressed adequately and I have no further comments on the study. I would like to thank the authors for their efforts in carrying out a thorough revision, and in particular for incorporating the suggested changes to their analysis.